# The genome of broomcorn millet

Changsong Zou [1,2], Leiting Li [1], Daisuke Miki[1], Delin Li[3,4,5], Qiming Tang[1], Lihong Xiao[1], Santosh Rajput[4], Ping Deng[1], Li Peng[1], Wei Jia[1], Ru Huang[1], Meiling Zhang[1], Yidan Sun[1], Jiamin Hu[1], Xing Fu[1], Patrick S. Schnable [3,4,5,6], Yuxiao Chang [7], Feng Li[1], Hui Zhang[8], Baili Feng[9], Xinguang Zhu[10], Renyi Liu[1], James C. Schnable [3,4,11], Jian-Kang Zhu[1,12] & Heng Zhang [1,13]

Broomcorn millet (*Panicum miliaceum* L.) is the most water-efficient cereal and one of the earliest domesticated plants. Here we report its high-quality, chromosome-scale genome assembly using a combination of short-read sequencing, single-molecule real-time sequencing, Hi-C, and a high-density genetic map. Phylogenetic analyses reveal two sets of homologous chromosomes that may have merged ~5.6 million years ago, both of which exhibit strong synteny with other grass species. Broomcorn millet contains 55,930 protein-coding genes and 339 microRNA genes. We find Paniceae-specific expansion in several subfamilies of the BTB (broad complex/tramtrack/bric-a-brac) subunit of ubiquitin E3 ligases, suggesting enhanced regulation of protein dynamics may have contributed to the evolution of broomcorn millet. In addition, we identify the coexistence of all three $C_4$ subtypes of carbon fixation candidate genes. The genome sequence is a valuable resource for breeders and will provide the foundation for studying the exceptional stress tolerance as well as $C_4$ biology.

[1] Shanghai Center for Plant Stress Biology and CAS Center for Excellence in Molecular Plant Sciences, Chinese Academy of Sciences, 3888 Chenhua Rd, 201602 Shanghai, China. [2] Key Laboratory of Plant Stress Biology, State Key Laboratory of Cotton Biology, School of Life Sciences, Henan University, 85 Minglun Street, 475001 Kaifeng, Henan, China. [3] Data2Bio LLC, Ames, IA 50011-3650, USA. [4] Dryland Genetics LLC, Ames, IA 50010, USA. [5] China Agricultural University, 100193 Beijing, China. [6] Department of Agronomy, Iowa State University, Ames, IA 50011-3650, USA. [7] Agricultural Genomes Institute at Shenzhen, Chinese Academy of Agricultural Sciences, 518120 Shenzhen, China. [8] Key Laboratory of Plant Stress Research, Shandong Normal University, No. 88 Wenhua East Rd, Jinan 250014 Shandong, China. [9] School of Agronomy, Northwest Agriculture & Forestry University, 3 Weihui Rd, 712100 Yangling, China. [10] National Key Laboratory of Plant Molecular Genetics, CAS Center for Excellence in Molecular Plant Sciences, Shanghai Institute of Plant Physiology and Ecology, Chinese Academy of Sciences, 300 Fenglin Rd, 200032 Shanghai, China. [11] Department of Agriculture and Horticulture, University of Nebraska-Lincoln, Lincoln, NE 68588, USA. [12] Department of Horticulture and Landscape Architecture, Purdue University, West Lafayette, IN 47907, USA. [13] National Key Laboratory of Plant Molecular Genetics, CAS Center for Excellence in Molecular Plant Sciences, Chinese Academy of Sciences, 3888 Chenhua Rd, 201602 Shanghai, China. Correspondence and requests for materials should be addressed to J.-K.Z. (email: jkzhu@sibs.ac.cn) or to H.Z. (email: zhangheng@sibs.ac.cn)

Drought is the most prevalent environmental stress in agriculture and decreases the yield of major crops by 50–80%[1]. Broomcorn millet (*Panicum miliaceum* L.) is a highly drought-tolerant cereal that is widely cultivated in the semiarid regions of Asia, Europe, and other continents. Originating from Northern China, it is also one of the world's earliest domesticated crops[2,3]. Because of its long history of worldwide cultivation by many different cultures, broomcorn millet has many other names including common millet, proso millet, and hog millet[4]. The grains of broomcorn millet are gluten-free and highly nutritious, containing higher contents of protein, several minerals, and antioxidants than most other cereals (Supplementary Figure 1)[5]. Broomcorn millet is therefore considered to be a crop that can potentially help ensure food security and diversify agriculture, and provide a healthier diet in the future[6].

Among all cultivated cereals, broomcorn millet has the highest water use efficiency (WUE, harvestable yield per water-use), probably because of its low respiration rate, short life cycle (60–90 days), and high harvest index[7,8]. It is mainly used for dryland farming where most other crops have failed, or as a summer rotation crop in temperate regions[6,9]. Broomcorn millet breeding programs have to date benefitted little from genomics technologies and have been conducted only on a small scale and in isolated regions of the world[9]. Only a small number of genetic markers and one genetic map have been published for broomcorn millet[10–15].

Broomcorn millet performs $C_4$ photosynthesis and is a close relative of the bioenergy crop switchgrass (*Panicum virgatum*). $C_4$ plants are more efficient in carbon fixation and in the use of water and nitrogen compared to their $C_3$ relatives. Thus much effort has been made in engineering $C_4$ traits in $C_3$ crops such as rice. This requires a clear understanding of the molecular mechanism of $C_4$ carbon fixation. $C_4$ plants are traditionally classified into three subtypes based on the main decarboxylation enzyme: nicotinamide adenine dinucleotide-dependent malic enzyme (NAD-ME), nicotinamide adenine dinucleotide phosphate-dependent malic enzyme (NADP-ME), or phosphoenolpyruvate carboxykinase (PEP-CK). However, multiple lines of evidence and mathematical modeling suggest the PEP-CK subtype does not operate independently and usually coexist as a supplemental pathway to either the NAD-ME or NADP-ME subtype[16,17]. NAD-ME $C_4$ grasses have higher WUE than their NADP-ME relatives[18]. *Panicum* is traditionally classified as the typical NAD-ME subtype, while the closely related *Setaria* mainly use NADP-ME. Different models have been proposed to explain the divergence of the two $C_4$ subtypes in *Panicum* and *Setaria*[8,17].

In addition to being nutritious and water-use efficient, broomcorn millet is also important for understanding the origin of agriculture in Eurasia. Nomadic farmers adopted broomcorn millet as a crop 8000–10,000 years before the present (BP) on the Loess Plateau of Northern China, where agriculture in East Asia originated[2,3,19]. Until ~3000 years BP, broomcorn millet, together with foxtail millet (*Setaria italica*), was intensively cultivated as the staple crop in Northern China[19]. By ~3000 years BP, broomcorn millet had spread across Europe and other parts of the continent through trade routes along the mountain valleys of Central Eurasia[19]. A survey of 98 landrace accessions of broomcorn millet using simple sequence repeat (SSR) markers indicated that genetic diversity within this species is closely associated with its geographical origin, possibly reflecting its history of spread across the continent[12].

In this study, we report a high-quality assembly covering 91.9% of the predicted nuclear genome, 94.2% of which was assigned to 18 pseudochromosomes. Phylogenetic analyses indicate that the two ancestral genomes of this allotetraploid diverged ~5.6 million years ago (MYA). We find lineage-specific expansion of an ubiquitin E3 ligase subunit in the genome, which may have contributed to its adaptive evolution. In addition, we identify $C_4$ candidate genes belonging to all three $C_4$ subtypes in broomcorn millet, suggesting that three different carbon fixation pathways may coexist in this plant.

## Results

**Genome sequencing and assembly.** A broomcorn millet landrace originating from Northern China was selected for genome sequencing and assembly. The genome size was estimated to be ~923 Mb based on a K-mer analysis (Supplementary Figure 2); this value is consistent with the reported c-value for this species[20]. Broomcorn millet is an allotetraploid with 36 chromosomes ($2n = 4\times = 36$)[21]. An 87-fold coverage of PacBio sequencing data were assembled using Canu[22] and error-corrected with PCR-free Illumina reads to reach an estimated consensus error rate of ~0.004% (1 error per ~25 kb), generating Pm_0390_v0.1 that contains 5541 contigs and has a contig N50 of 369 kb (Supplementary Tables 1 and 2, Supplementary Figure 3). We sequenced 132 individuals from an F6 population of recombinant inbred lines (RIL) at an average depth of ~10-fold and constructed a genetic map consisting of 18 linkage groups (LG) and 221,787 single nucleotide polymorphism (SNP) markers (Supplementary Figure 4). We anchored 4146 contigs to this high-density genetic map. We also arranged the contigs into 18 groups based on the spatial relationship deduced from an Hi-C assay. By combining the position information from the genetic map and the Hi-C experiment, we assembled 4250 contigs into 18 pseudochromosomes with a total length of 822 Mb (Supplementary Table 3). The resulting final assembly (Pm_0390_v1) contains 18 pseudochromosomes and 1291 unassigned contigs, covering 92.6% (855 Mb) of the estimated nuclear genome with 1.98% undetermined bases (Table 1, Supplementary Table 3). The 18 pseudochromosomes range from 32.2 to 66.9 Mb and are numbered in descending order of their lengths. The total length of the pseudochromosomes accounts for 96.1% (822 Mb) of the assembly (Fig. 1, Supplementary Table 3). By filtering reads that show sequence similarity to known chloroplast genomes, we were able to assemble the chloroplast genome into a single contig with a length of 140,048 bp. Further annotation identified 116 genes in this organellar genome (Supplementary Figure 5).

**Table 1 Global statistics of *P. miliaceum* genome assembly and annotation**

|                              | Number         | Size       |
|------------------------------|----------------|------------|
| Assembly feature             |                |            |
| Estimated genome size        |                | 923 Mb     |
| Total scaffolds (≥1000 bp)   | 1,309          | 855 Mb     |
| Undetermined bases           | 1.98%          | 16.9 Mb    |
| Scaffold N50                 | 8              | 46,662 kb  |
| Longest scaffold             |                | 66,885 kb  |
| Pseudochromosomes            | 18             | 822 Mb     |
| Anchored contigs             | 4,146          | 805 Mb     |
| Anchored and oriented contigs| 3,242          | 722 Mb     |
| Total contigs                | 5,541          | 838 Mb     |
| Contig N50                   | 423            | 369 kb     |
| Longest contig               |                | 5222 kb    |
| GC content                   | 46.8%          |            |
| Genome annotation            |                |            |
| Repetitive sequences         | 58.2%          | 495 Mb     |
| Protein-coding genes         | 55,930         | 181 Mb     |
| Genes in pseudochromosomes   | 55,527 (99.3%) |            |
| Noncoding RNAs               | 9643           | 1.5 Mb     |

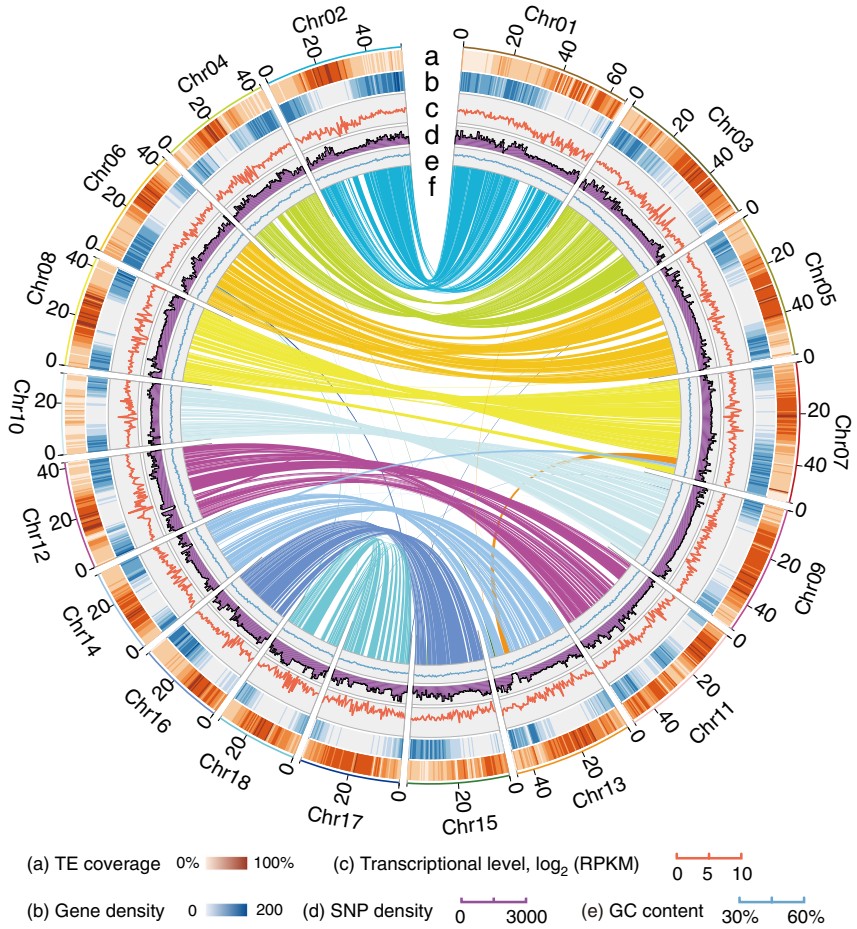

**Fig. 1** Synteny and distribution of features in the broomcorn millet genome. The number and length (Mb) of pseudochromosomes are indicated outside of the ring. **a** TE coverage, **b** gene density, **c** average transcript levels, **d** marker density represented by the number of SNPs and **e** GC (guanine-cytosine) content of the genome in 1-Mb nonoverlapping windows. **f** Synteny blocks >1 Mb long among homologous broomcorn millet chromosomes are indicated. TE transposable elements, SNP single nucleotide polymorphism .

To assess the quality of Pm_0390_v1, a fosmid library was generated, and ten colonies were randomly selected for PacBio long-read sequencing. At a mean coverage of ~1000-fold, 10 contigs ranging from 24 to 46 kb in length (average length ~35 kb) were de novo assembled. Alignment of the fosmid sequences to Pm_0390_v1 revealed no structural errors and high sequence identity rates (99.53–100%) (Supplementary Table 3). The coverage of gene space in Pm_0390_v1 was estimated using transcriptome data from eight different broomcorn millet tissues (Supplementary Table 1). A total of 305,520 transcripts were de novo assembled from 241 Gb of mRNA-seq data. More than 98% of the transcript sequences could be mapped to Pm_0390_v1 (Supplementary Table 5), indicating excellent coverage of expressed genes. In addition, 1411 (98%) of the 1440 plant single-copy orthologs from BUSCO v2[23] were identified in the broomcorn millet genome (Supplementary Table 6). These metrics indicate that the assembly has high accuracy and completeness.

**Genome annotation**. We annotated repetitive sequences of the genome using both in silico prediction and homology-based approaches. The integrated results indicated that the broomcorn millet genome has a repeat content of 58.2% (Table 1), of which 92.1% consists of transposable elements (TE). As observed in many plant genomes, most TE sequences (406 Mb) are retro-transposons (Class I TE), with the Gypsy and Copia superfamilies

being the dominant types (82.9% of TEs) (Supplementary Table 7). We also identified 112,158 SSR with a mean occurrence frequency of 22.5 per Mb (Supplementary Table 8). Most of the SSRs were composed of di- and tri-nucleotide motifs with an average length of ~22 bp (Supplementary Table 9). The results of the analysis can serve as a resource for developing SSR-based genetic markers.

To produce accurate gene models and to obtain a global picture of gene expression during broomcorn millet development, we generated 241 Gb of mRNA-seq data for eight representative types of tissues (Supplementary Table 1). The reads were aligned to the genome and assembled into transcripts. After the transcriptome assembly and the results from ab initio prediction and homology search were integrated (Supplementary Table 10), we identified 55,930 protein-coding genes and 339 microRNA (miRNA) genes, in addition to 1420 transfer RNAs, 1640 ribosomal RNAs, and 2302 small nuclear RNAs (Supplementary Table 11). The 18 pseudochromosomes contain 55,527 (99.3%) protein-coding genes (Table 1). On average, protein-coding genes in broomcorn millet are 3260 bp long and contain 4.7 exons (Supplementary Table 10), and these values are similar to those of other monocotyledonous species. We assigned functions to 96.6% (54,003) of the protein-coding genes based on sequence similarity (Supplementary Table 12).

About 70% of the predicted gene models had a probabilistic confidence score of 1.0, indicating high concordance of results generated by different approaches (Supplementary Figure 6)[24].

More than 73% of the gene models contained conserved domains listed in the Pfam database[25].

**Evolutionary history of broomcorn millet.** For comparative analyses, we selected six other grass species whose genomes have been sequenced: three were from the PACMAD clade (*S. italica* = foxtail millet, *Zea mays* = maize, and *Sorghum bicolor* = sorghum), and three were from the BEP clade (*Triticum aestivum* = wheat, *Brachypodium distachyon* = stiff brome, and *Oryza sativa* = rice). A phylogenetic analysis using 511 single-copy orthologous genes confirmed the close relationship between *S. italica* and *P. miliaceum* (Paniceae tribe) and between *Z. mays* and *S. bicolor* (Andropogoneae tribe) (Fig. 2a). Using a reference divergence time of 32–39 MYA between *Brachypodium* and wheat and 40–53 MYA between *Brachypodium* and rice[26], we estimated that *Setaria* and *Panicum* shared a common ancestor ~18 MYA (Fig. 2a).

We identified homologous gene pairs in broomcorn millet and foxtail millet and estimated species divergence time using fourfold degenerate transversion (D4DTv) distance. All gene pairs showed a shallow peak at 0.38, likely reflecting the rho (ρ) whole genome duplication (WGD) event that occurred ~70 MYA in the grass lineage[27] (Fig. 2b). Paralogous gene pairs of Pm peaked at 0.032, and Pm-Si gene pairs peaked at 0.081 (Fig. 2b). These numbers suggest that the tetraploidization of broomcorn millet occurred ~5.8 MYA.

Synteny was detected across the genome, both among the chromosomes of the allotetraploid and between broomcorn millet and other species. A large number of synteny blocks exist between pairs of Pm chromosomes (Fig. 1), which is consistent with the predicted hybridization between two closely related *Panicum* species. In total, 604 synteny blocks with an average size of 1.31 Mb were identified in broomcorn millet by comparing to the foxtail millet genome. For each synteny block in foxtail millet, usually two were identified in broomcorn millet, and were mostly located on separate chromosomes (Fig. 2c). A smaller number (525) of syntenic regions were identified in broomcorn millet when compared to sorghum (*Sb*). Based on these analyses, orthologous relationships between chromosomes were identified. For example, Chr1 and Chr2 of broomcorn millet share origins with Chr9 of foxtail millet and with Chr1 of sorghum (Fig. 2c, Supplementary Figure 7). Chromosome-scale rearrangements were also observed. For example, broomcorn millet Chr5 and Chr6 seems to correspond to a fusion between part of Chr8 and Chr9 from sorghum (Fig. 2c, Supplementary Figure 7).

**Comparative genomics of gene families.** Based on sequence homology, we assigned 47,142 broomcorn millet genes to 20,374 families. Relative to the most recent common ancestor of broomcorn millet and foxtail millet, expansion in over half of the gene families (11,773 of 20,374) was observed (Fig. 3a).

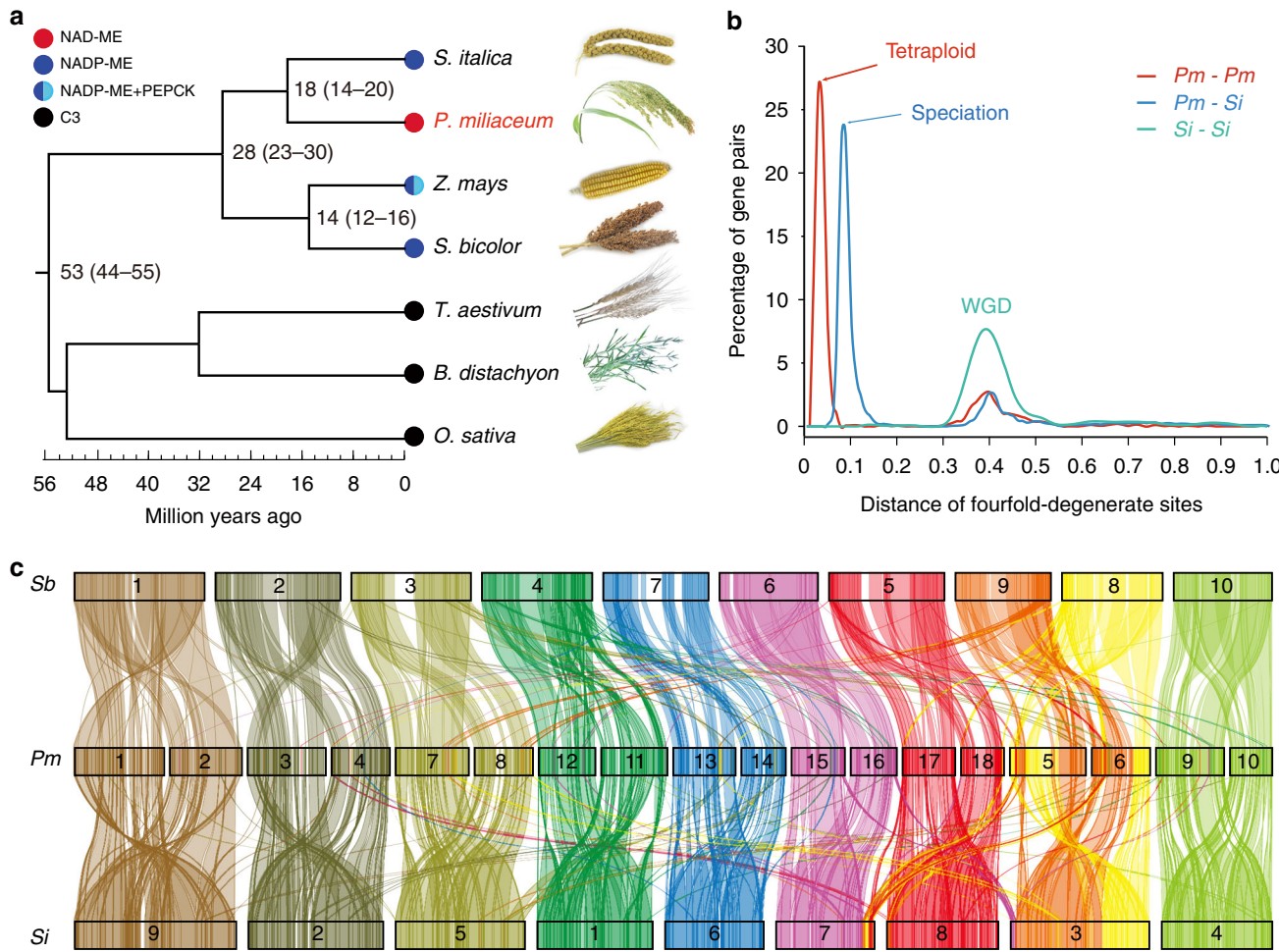

**Fig. 2** Evolutionary analyses of the broomcorn millet genome. **a** Species phylogenetic tree constructed from single-copy orthologs. Lineage divergence time is indicated at each branch point. The photosynthesis type of each species is indicated by colored dots at each node. **b** 4DTv distance of homologs genes from broomcorn millet (*Pm*) and foxtail millet (*Si*). **c** Synteny blocks identified between broomcorn millet and foxtail millet, and between broomcorn millet and sorghum (*Sb*). Only synteny blocks >0.5 Mb long are shown.

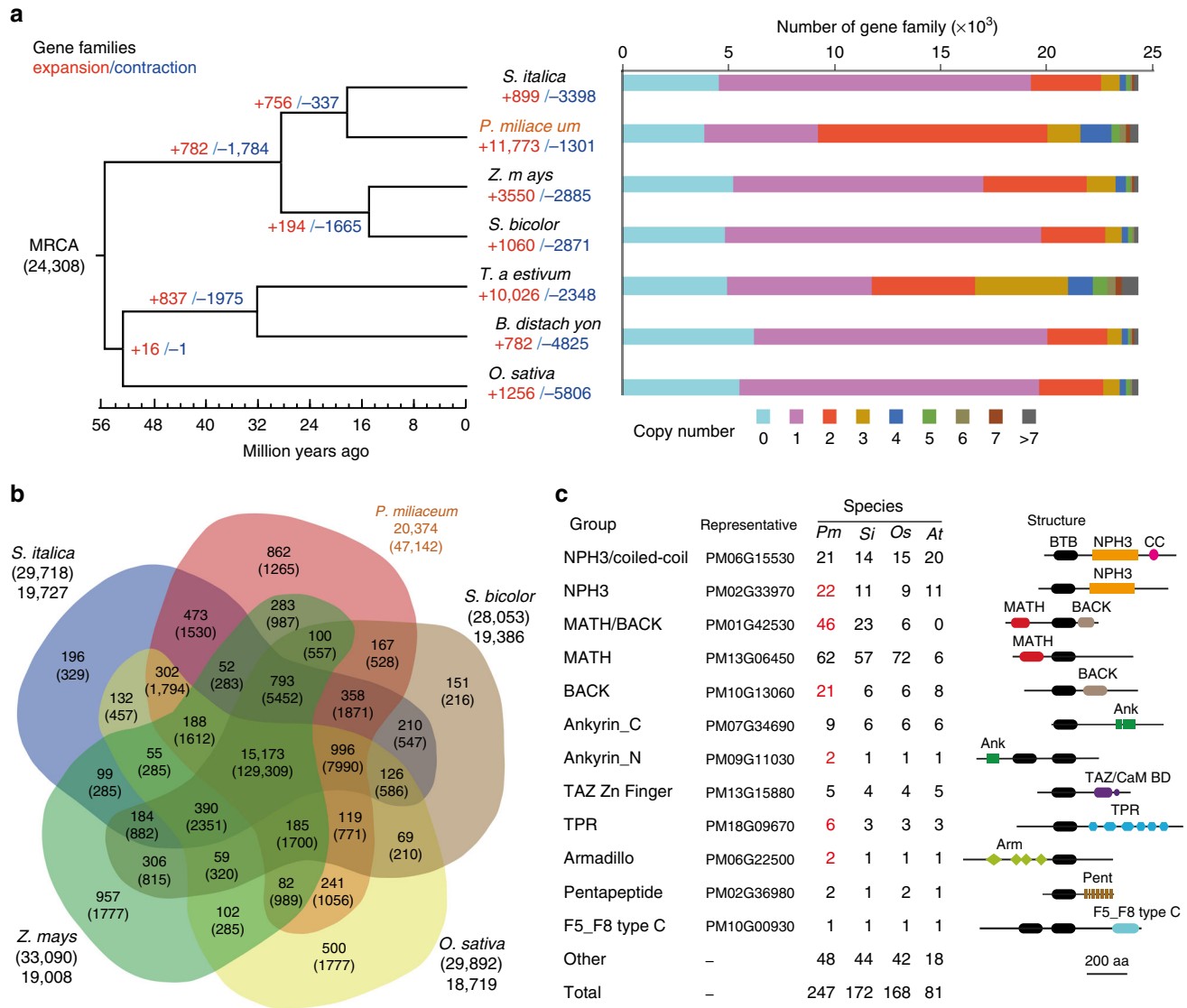

**Fig. 3** Comparative genomics of gene families in broomcorn millet. **a** The number of gene families that expanded or contracted during evolution mapped to the species phylogenetic tree. **b** Overlap of gene families in broomcorn millet (*P. miliaceum*) and four other grasses. **c** Gene copy number and domain architecture of BTB proteins in broomcorn millet (*Pm*), foxtail millet (*Si*), rice (*Os*), and *Arabidopsis thaliana* (At). Gene copy numbers that are at least twofold higher in broomcorn millet (*P. miliaceum*) than in other species are labeled red. BTB broad complex/tramtrack/bric-a-brac, NPH3 nonphototropic-hypocotyl 3, MATH meprin-and-TRAF-homology, BACK BTB and C-terminal Kelch, TAZ Transcription Adaptor putative Zinc finger, TPR Tetratricopeptide repeat, F5_F8 type C discoidin domain.

Expansion in a similar number of gene families (10,026) was also observed for wheat, a hexaploid crop. Of the broomcorn millet gene families, 52.9% contain two copies and 19.8 and 18.0% of wheat gene families contain two and three copies respectively (Fig. 3a). Most of the two-copy gene families of broomcorn millet are located in synteny blocks, indicating that the expansion was mainly due to the recent WGD event (Supplementary Figure 8). We examined the relationship between the size of gene families and gene expression pattern in different broomcorn millet tissues. Gene families with higher copy numbers tended to have lower Shanon Entropy values and therefore more tissue-specific expression patterns, indicating diversified expression patterns among their members (Supplementary Figure 9A). Interestingly, two-copy gene families had the most uniform expression and highest average expression levels (Supplementary Figure 9B). Gene Ontology (GO) enrichment analysis identified genes involved in protein binding, nucleic acid binding, and ion binding in two-copy gene families (Supplementary Figure 9C).

By comparing broomcorn millet with four other grasses including foxtail millet, maize, sorghum, and rice, we found that 74.5% (15,173/20,374) of the gene families in broomcorn millet were shared among all five species, while only 4.2% (862) of the gene families were specific to broomcorn millet (Fig. 3b). Among the broomcorn millet-specific families, more than 300 were predicted to encode nuclear proteins, and genes involved in protein phosphorylation and protein–protein interactions were significantly over-represented (Supplementary Figure 10). Of the 1313 orthologous groups encoding transcription factors (TFs), 899 were expanded relative to the ancestor. This portion was significantly higher than average level of expansion in the genome ($p = 3.3e{-}14$, two-tailed proportion test). HSF (heat shock factor) is one of a few TF families that did not expand, while the three TF families in broomcorn millet that had expanded the most were SRS (SHI related sequence), EIL (EIN3-like), and C3H (Cys3-His zinc finger) (Supplementary Table 13).

Among the gene orthologous groups in broomcorn millet that expanded the most were ubiquitin E3 ligase subunits. Further investigation indicated that they all contained the BTB domain, which forms a complex with CUL3 (cullin-3) and RBX1 (RING-box protein 1) and is involved in target recognition of the E3 ligase[28]. We therefore performed a comprehensive analysis of the BTB proteins in broomcorn millet and compared them to those in foxtail millet, rice, and *Arabidopsis*. The number of BTB proteins in the grass family was significantly higher than in *Arabidopsis* and was highest in broomcorn millet (Fig. 3c). Based on the identity and arrangement of other domains, we divided the BTB proteins into subgroups (Fig. 3c). Broomcorn millet contains more copies in most subgroups than the other three species. Consistent with a previous study in rice[29], the MATH-BTB subgroup is strongly expanded in grass species compared to *Arabidopsis*. The MATH-BTB-BACK subgroup is strongly expanded in *Panicum* and *Setaria*, but contains six copies in rice and 0 copies in *Arabidopsis* (Fig. 3c). The expansion of BTB-BACK proteins (21 copies in broomcorn millet) was specific to *Panicum* because the subgroup in the other species contained only 6–8 copies (Fig. 3c). A phylogenetic tree constructed from BTB domain sequences of broomcorn millet or foxtail millet indicated that the clustering was largely consistent with the classification based on domain architecture. The MATH-BTB, MATH-BTB-BACK, and BTB-BACK proteins were clustered into a clade different from the other subgroups, while the expansion of BTB-BACK and MATH-BTB-BACK was not restricted to a single branch of the tree (Supplementary Figure 11), indicating that the duplication of BTB genes in the Paniceae occurred multiple times.

**Genes involved in C$_4$ photosynthesis.** C$_4$ plants typically have higher water-use efficiency than plants performing C$_3$ carbon fixation, conferring them a competitive advantage in arid and semiarid regions[30]. We thus analyzed the evolution and expression of C$_4$-related genes in broomcorn millet. In the classic NAD-ME model, aspartate (Asp), derived from oxaloacetate (OAA), is the main metabolite transported from M cells to BS cells; in the mitochondria of BS cells, Asp is converted to OAA and then malate, which is decarboxylated by NAD-ME (Fig. 4a). Evidences also suggest that OAA could be decarboxylated in BS cytosol by phosphoenolpyruvate carboxykinase (PEPCK). This process functions as a supplement to the classic NAD-ME model[16,17].

We analyzed the copy number of genes involved in C$_4$ carbon fixation, including enzymes and metabolite transporters, and found that except for dicarboxylate transport 2 (DiT2) and mitochondrial pyruvate carrier all of them have a higher copy number in broomcorn millet than in foxtail millet, the ratio of which was usually twofold (Supplementary Table 14). These genes were also located in syntenic regions that are conserved within the grass family. For example, we identified eight copies of carbonic anhydrase (CA), four copies of NAD-ME and eight copies of NADP-ME in broomcorn millet. All of these genes in broomcorn millet were syntenic with their orthologs in foxtail millet, sorghum, or rice (Fig. 4b–d). Each synteny block from these diploid species corresponded to two blocks in broomcorn millet, each located on two homologous chromosomes (Fig. 4b–d).

We identified candidate enzymes involved in C$_4$ carbon fixation in broomcorn millet based on their preferential expression in photosynthetic tissues. All the enzymes characterizing the NAD-ME subtype, including NAD-ME, NAD-MDH, AspAT, and AlaAT, were identified (Fig. 4e). For example, the transcript levels of two candidate NAD-MEs (PM01G38550 and PM02G10170) were over 1250- and 43-fold higher in leaf blades than in roots and seeds (Fig. 4e and Supplementary Figure 12).

We also found that the proteins specific for the NADP-ME subtype, such as NADP-ME and NADP-MDH, were more highly expressed in photosynthetic tissues (Fig. 4e and Supplementary Figure 12). Two NADP-ME genes (PM07G37230 and PM08G02950) were expressed at similar levels as the two C$_4$ NAD-MEs in leaf tissues. Their expression levels in 1-week-old seedlings were even higher than the two C$_4$ NAD-MEs (Fig. 4e). These results suggest a mixed C$_4$ model that contains features from the traditional NAD-ME and NADP-ME subtypes in broomcorn millet (Fig. 4a). The candidate C$_4$ metabolite transporters were consistent with this model. We identified not only the mitochondria-localized malate phosphate antiport 1 (DIC1)[31], but also the coupled bile acid sodium symporter 2 and sodium:hydrogen antiporter (BASS2/NHD)[32] and dicarboxylate transporter 2 (DiT2), which were presumably localized to the chloroplast (Fig. 4e).

We further performed phylogenetic analyses on C$_4$-related genes using the coding sequences from six grass species and *Arabidopsis thaliana* (Supplementary Figure 13). The results indicated that all the C$_4$ candidate genes come from clades that contain C$_4$ genes of other grasses[33]. The NAD-MEs contain two lineages that diverged early in angiosperm evolution (Supplementary Figure 13a). The two clades each contain the α and β subunit of NAD-ME from *Arabidopsis*[34]. The two candidate C$_4$ NAD-MEs from broomcorn millet belong to group 2 (Supplementary Figure 13a). Biochemical purification of leaf NAD-MEs from *Panicum dichotomiflorum* indicated that they mainly exist as homo-octamers[35], supporting the inference that only group 2 NAD-MEs are used for C$_4$ photosynthesis in broomcorn millet. Similar analyses indicated that NADP-MEs diverged early and had different lineages in monocots and dicots (Supplementary Figure 13b). The NADP-MEs in monocots can be divided into four clades. Group 4 contains all of the C$_4$ NADP-MEs from foxtail millet and maize[33,36] (Supplementary Figure 13b), as well as the two NADP-MEs from broomcorn millet that were preferentially expressed in seedlings and other photosynthetic tissues (Fig. 4e).

## Discussion

Climate change has far-reaching and adverse effects on crop yields and human nutrition[37]. To make matters worse, an increasing world population will require that current food production be doubled by the year 2050[38]. In addition, farming land is being lost to urbanization, soil deterioration, and extreme weather. Responding to these problems will require the development of stress-tolerant crops. Although much progress has been made in understanding the response to drought stress in several model plants, the development of transgenic crops that are drought-tolerant has so far been difficult[39]. Broomcorn millet consumes less water and is more drought-tolerant and nutritious than most other cereals. Although the land area planted with broomcorn millet has been declining due to the planting of crops with higher yields[9], a record broomcorn millet yield of 4500 kg/ha was achieved in Fugu, China (Feng BL, unpublished results). This indicates that the potential for increasing broomcorn millet yield is substantial. Much of what we have learned about increasing the yield of rice and other main cereals can be readily applied to broomcorn millet[39] (Supplementary Figure 1). The genetic diversity of broomcorn millet varieties from different regions of the world remains a valuable but unexplored resource. Broomcorn millet could be used not only as a dryland crop but also as a crop in broader regions to support more water-efficient, sustainable agriculture. The genome assembly from the current study provides a foundation for the molecular breeding of broomcorn millet.

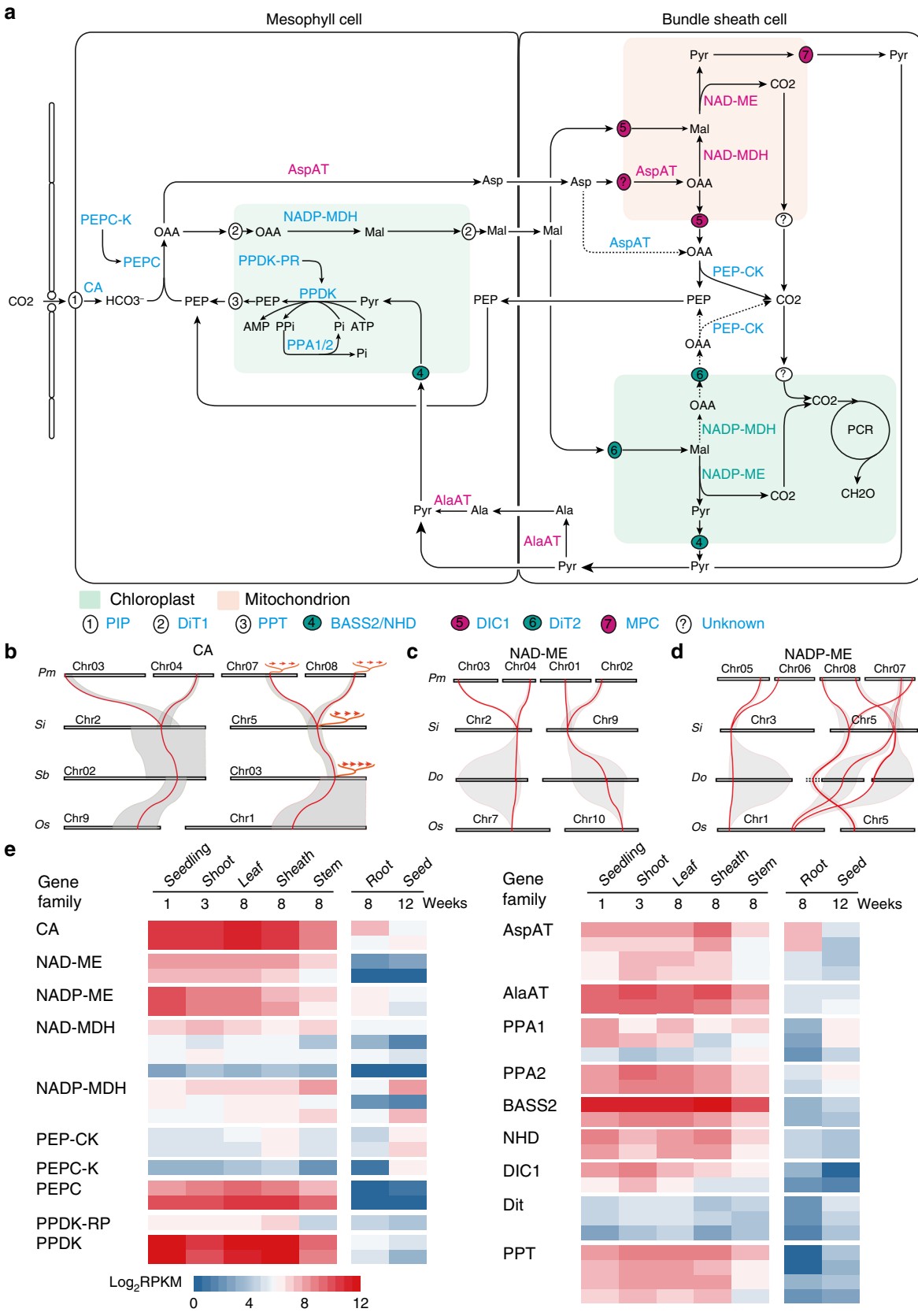

Synteny and gene family analyses in broomcorn millet have provided important clues regarding its evolution. We showed that the broomcorn millet genome resulted from hybridization between two closely related genomes ~5.6 MYA. Most *Panicum* species are polyploid[40] and are native to tropical/semiarid regions of the world. A large proportion of gene families in broomcorn millet genome are two-copy, most of which were retained from single-copy genes of both parental species (Fig. 3a). Genes

**Fig. 4** A proposed model of C$_4$ photosynthesis in broomcorn millet. **a** Diagram depicting the main proteins and metabolic fluxes involved in C$_4$ photosynthesis. Proteins are colored based on traditional models: the ones that commonly function in NAD-ME and NADP-ME C$_4$ are in blue; the ones that participate in NAD-ME C$_4$ are in magenta; the ones that participate in NADP-ME C$_4$ are in green. Abbreviations for metabolites and enzymes: CO$_2$ carbon dioxide, Ala alanine, Asp aspartate, Mal malate, Pyr pyruvate, OAA oxaloacetate, PEP phosphoenolpyruvate, CA carbonic anhydrase, PEPC phosphoenolpyruvate carboxylase, PPDK pyruvate/orthophosphate dikinase, AspAT aspartate aminotransferase, AlaAT alanine aminotransferase, NADP-MDH NADP-dependent malate dehydrogenase, NADP-ME NADP-dependent malic enzyme, NAD-MDH NAD-dependent malate dehydrogenase, NAD-ME NAD-dependent malic enzyme, PEPC-K PEPC kinase, PEP-CK phosphoenolpyruvate carboxykinase. Metabolite transporters are presented by circled numbers or question marks on the membrane: 1, plasma membrane intrinsic protein (PIP); 2, dicarboxylate transporter 1 (DiT1); 3, phosphate/phosphoenolpyruvate translocator (PPT); 4, sodium bile acid symporter 2 (BASS2) and sodium: hydrogen antiporter (NHD); 5, malate phosphate antiport 1 (DIC1); 6, dicarboxylate transporter 2 (DiT2); 7, Mitochondrial pyruvate carrier. **b**–**d** Synteny blocks containing **b** CA, **c** NAD-ME and **d** NADP-ME genes. Chromosomes or scaffolds are shown as rectangular boxes, which are not drawn to scale. Synteny blocks are shaded gray with red lines indicating the position of genes. The species are labeled as follows: Pm = broomcorn millet, Si = foxtail millet, Sb = sorghum, Do = *Dichanthelium oligosanthes*, Os = rice. **e** Heatmap showing the expression level of candidate genes involved in C$_4$ carbon fixation in photosynthetic and nonphotosynthetic tissues of broomcorn millet.

in two-copy families exhibit more uniform and higher expression levels than other-size gene families. They are also enriched in genes involved in nucleic acid binding and protein−protein interactions (Supplementary Figure 9). Similar biased retention of gene families encoding subunits of protein complexes have been reported in other species after WGD[41]. Since polyploids were shown to exhibit increased drought tolerance in several plant species[42], it will be interesting to test in the future whether similar classes of genes were retained in other polyploid *Panicum* species, which could contribute to their adaptive evolution.

We also observed lineage-specific expansion in the BTB protein family, a subunit of ubiquitin E3 ligase (Fig. 3c). The ubiquitin-proteasome system (UPS) is involved in many aspects of plant hormone signaling and stress responses. In particular, many components of the ABA signaling pathway are regulated by ubiquitination and proteasome degradation[43]. Ubiquitin-like proteins including ATGs play roles in autophagy and could be important for efficient utilization of nutrients under stress conditions[44]. The expansion of BTB-MATH proteins seems to be specific for *Panicum*, while the expansion of MATH-BTB-BACK is restricted to *Panicum* and *Setaria*. Both MATH and BACK domains are involved in protein–protein interactions. Because BTB proteins are involved in target recognition of ubiquitin E3 ligases, the recruitment of unique domains and amplification in specific gene families suggests that diversified protein targets regulated by UPS in broomcorn millet may be importation for its adaptation.

The elucidation of the broomcorn millet genome has also provided unique insights into the evolution and metabolic pathways of NAD-ME type C$_4$ plants. We identified a number of C$_4$ candidate genes in broomcorn millet based on the analyses of expression levels, synteny, and phylogenetic relationships of genes. The assignment of genes to C$_4$-related functions was consistent with biochemical studies. For example, purification of AspATs from broomcorn millet leaves identified two main AspATs involved in C$_4$ carbon fixation: the cytosolic AspAT from M cells and the mitochondrial AspAT from BS cells[45,46]. Both forms of C$_4$ AspAT genes (two copies each) were identified in our analyses (Fig. 4e). Our further analyses indicated that both NAD-ME and NADP-ME subtype-related enzymes were more highly expressed in photosynthetic tissues of broomcorn millet (Fig. 4e and Supplementary Figure 13). Consistent with the existence of NADP-ME decarboxylation in the chloroplast of BS cells, we also identified highly expressed pyruvate transporters (BASS2/NHD) and putative malate transporters (DiT2), both of which can be localized to the chloroplast and promote the import of malate and export of pyruvate or oxaloacetic acid. All these suggest that these three different decarboxylation mechanisms can potentially coexist in one single C$_4$ species; utilizing more

than one decarboxylation mechanisms can help cope with dynamic and fluctuating environments in the field[16,47].

## Methods

**Plant materials and growth conditions**. The broomcorn millet landrace (accession number 00000390) sequenced in this study was ordered from National Crop Germplasm Resources Reservation Center of China. The landrace was originally collected from Antu, Jilin Province, China. Plants were grown in a temperature (27 ± 1 °C) and humidity (45–55%) controlled growth room with 900 μmol/m$^2$/s (measured 40 cm beneath the light) light intensity and 14h−10h day−night cycle. Before using for sequencing, plants were propagated for three generations through self-pollination.

**Genome sequencing**. The leaf tissue from 3-week-old plants was collected and flash-frozen in liquid nitrogen. Genomic DNA was extracted from the leaf tissue using DNeasy Plant Maxi kit (Qiagen). For the Illumina PCR-free library, genomic DNA was fragmented in a Covaris S220 and separated on a SAGE-ELF (Sage Science) following the manufacturer's instructions. The fraction that is 310~450 bp in size from SAGE-ELF were used for PCR-free library construction using TruSeq Nano DNA Library Preparation Kit (Illumina). The 20-kb PacBio library were prepared and sequenced on PacBio RS II using P6-C4 chemistry at Tianjin Biochip Corporation, following the manufacturer's standard protocols.

Hi-C was performed following a published protocol[48]. Briefly, 2 g of 10-day-old broomcorn millet seedlings were fixed in 1% formaldehyde solution. The nuclei/chromatin was extracted from the fixed tissue and digested with *Hin*dIII (New England Biolabs). The overhangs resulting from *Hin*dIII digestion were filled in by biotin-14-dCTP (Invitrogen) and the Klenow enzyme (NEB). After dilution and re-ligation with T4 DNA ligase (NEB), genomic DNA was extracted and sheared to a size of 300−500 bp with Bioruptor (Diagenode). The biotin-labeled DNA fragments were enriched using streptavidin beads (Invitrogen) and subject to library preparation.

**Construction of the genetic map**. A set of 132 RILs (F6) obtained from a bi-parental cross and the two parents (an Asian and a North American inbred line) were genotyped using whole genome sequencing. In total, 222,081 high-quality bi-allelic SNPs were called using bcftools (v1.7)[49] with the following criteria: (a) the missing data rate in progenies is less than 20%; (b) the segregation ratio fitting the predicted 31:2:31 (homozygotes as the first parent:heterozygotes:homozygotes as the second parent) has a P value higher than 1e-5 by chi-squared test; (c) variation quality ≥999. Lep-MAP3 (v0.2)[50] was used for genetic map construction. A LOD (logarithm of odds) score of 13 and a fixed recombination fraction of 0.03 were used for separating different LG. A total of 18 LG each containing at least 3525 were identified. The order of markers and the genetic distance were then estimated using the Kosambi mapping function. The final genetic map included 221,787 SNP markers and a total genetic length of 2811 cm for the maternal parent and 3092 cm for the paternal parent.

*Genome assembly*: Filtered subreads (81.03 Gb) were used for assembly with canu (v1.7)[22] given genomeSize parameter as 900 M, and errorRate was set to 0.013 to improve assembly quality. Primary contigs were polished using Pilon (v1.22)[51] with PE250 PCR-free reads.

The reads from the Hi-C library were preprocessed (removing adapter sequences and low-quality bases) before being aligned to Pm_0390_v0.1 assembly using the aln and sampe commands from bwa (v0.7.17)[52]. The resulting bam files together with the contigs from Pm_0390_v0.1 assembly were used as input for LACHESIS (https://github.com/shendurelab/LACHESIS)[53] with the cluster number set to 18 and other parameters as default. The Hi-C map was then converted to a 100-cm pseudo-map with two pseudo-markers per contig (Hi-C map). The Hi-C map, the genetic maps of two parents and the contig sequences

from Pm_0390_v0.1 were used as input for ALLMAPS (v0.8.4)[54] to generate 18 pseudochromosomes and 1292 unassigned contigs; the map weight for the Hi-C map was set to 1 and the map weight for the genetic maps was set to 10 in ALLMAPS. The serial numbers of chromosomes were manually adjusted to reflect the descending order of chromosome length (Chr01—longest; Chr18—shortest). This final assembly of 18 chromosomes and 1291 contigs was named Pm_0390_v1.

**Assessment of genome assembly.** PE250 (pair end 250 bp) reads from the PCR-free library were used to estimate the consensus error rate. The preprocessed (adaptor and low-quality bases trimmed) reads were aligned to Pm_0390_v1 using bwa mem[52] with default parameters. Then samtools (v 0.1.19) and GATK (v4.0.3.0; https://software.broadinstitute.org/gatk/) were used for SNP calling and summarization.

A 40-kb fosmid library for broomcorn millet was constructed using CopyControl Fosmid Library Production Kit (Epicenter). After the transformation, ten single colonies were picked and cultured in 100-mL LB medium. The ten fosmids were then extracted using a Plasmid Midi Kit (Qiagen), mixed in equal molar and used for the preparation of a 20-kb PacBio library. Library preparation and sequencing were performed at Tianjin Biochip Corporation. Falcon v0.3.0 with default parameters was used for the de novo assembly of fosmid sequences. After removing the original plasmid backbone, the contigs were then aligned to Pm_0390_v1 using blastn and the results were summarized manually.

The completeness of the assembly was assessed using both transcriptome data and BUSCO (Benchmarking Universal Single-Copy Orthologs)[23]. First raw mRNA-seq reads from eight types of broomcorn millet tissue (Supplementary Data 1) were trimmed and combined. Trinity[55] was used for de novo transcriptome assembly in the no reference mode. The 305,520 assembled transcripts were then compared to Pm_0390_v1 using blastn with default parameters (Supplementary Table 6). The results were summarized using an in-house Perl script. The 1440 embryophyta single-copy orthologs in BUSCO v2 was compared to Pm 0390v1 with a BLAST E value cutoff of 1e-5.

**Genome annotation.** Annotation of the chloroplast genome was performed separately using DOGMA (webtools, http://dogma.ccbb.utexas.edu) and CpGAVAS (http://www.herbalgenomics.org/0506/cpgavas) with the following parameters: blast E value cutoff—1e-10, maximum target hit number—10, and maximum length of tRNA intron and variable region—116 bp. Then outputs from the two software were integrated by retaining the longer opening read frame (ORF) with an in-house Perl script. The predicted start/stop codons and the exon—intron boundaries for intron-containing genes were manually examined and curated. The map of the chloroplast genome was generated using GenomeVx[56] followed by manual adjustment.

Both homology-based and de novo approaches were used for repeat annotation. Three complementary software programs, LTR_FINDER (v1.06)[57], PILER (v1.0)[58], RepeatModeler (v4.0.6)[59], were used to generate a de novo repeat library for broomcorn millet. Default parameters were used unless otherwise noted. This de novo repeat library was then used together with Repbase for homology search of repeats using RepeatMasker (v1.0.10)[60].

Three independent approaches, including ab initio prediction, homology search, and reference guided transcript assembly, were used for gene prediction in a repeat-masked genome. Evidence from the three approaches were then integrated using GLEAN (v1.0.1)[24] to generate the final gene set.

*Ab initio gene prediction:* AUGUSTUS (v2.5.5), Genescan (v1.0), SNAP (version 2006-07-28), GlimmerHMM (v3.0.3), and Fgenesh (http://www.softberry.com) with default parameters were utilized for ab initio gene prediction with parameters trained with the rice[61] and foxtail millet[62] gene models. Genes with CDS (coding sequence) less than 150 bp in length were discarded.

*Homology-based gene prediction:* Candidate ORFs in the broomcorn millet genome were identified by aligning the protein sequences of six grass species and *Arabidopsis thaliana* (Supplementary Table 9) to Pm_0390_v1 using TBLASTN with an E value cutoff of 1e−5. The candidate regions and the 2000-bp sequences upstream and downstream of them were extracted from the genome. Gene models were generated using GeneWise (v2.4.1) with parameters: -trev -sum -genesf from aligned protein sequences from other species to these DNA fragments.

*Transcriptome-assisted gene prediction:* TopHat (v2.1.1) was used to map filtered mRNA-seq reads to Pm_0390_v1 to identify exonic regions and intron −exon boundaries with the following parameters: -p 4 -max-intron-length 20,000-m 1 -r 20 -mate-std-dev 20. Cufflinks (v2.2.1) was then used to assemble the alignments into transcripts with the parameters: -I 20,000 -p 4.

**Functional annotation of gene models.** To assign gene functions, the predicted protein sequences were searched against six protein/function databases: InterPro, GO, KEGG, KOG, Swiss-Prot, and TrEMBL. The Interpro database search was performed using InterProScan with parameters: -f TSV –dp –gotermes -iprlookup –pa. For the other five databases, BLAST searches using the protein sequences as query were performed with an E value cutoff of 1e−05 and the results with the hit with lowest E value was retained. Results from the six database searches were concatenated. For GO term enrichment analysis, Fisher's exact test was performed and the *P* value was adjusted for multiple testing using the Benjamini−Hochberg method.

**Species phylogenetic analysis.** OrthoFinder (v1.1.4)[63] was used to identify the orthologous groups among seven grass species (*O. sativa, Z. mays, T. aestivum, S. italica, B. distachyon, S. bicolor, P. miliaceum*). All-versus-all BLASTP with an E value cutoff of 1e−05 were performed and orthologous genes were clustered using OrthoFinder. Single-copy orthologous genes were extracted from the clustering results. MAFFT v7 with default parameters was used to perform multiple alignment of protein sequences for each set of single-copy orthologous genes, and transform the protein sequence alignments into codon alignments. Poorly aligned or divergent regions were removed from the multiple sequence alignment results using Gblocks (v0.91b)[64] (minimum number of sequences for a conserved position: 9; minimum number of sequences for a flank position: 14; maximum number of contiguous nonconserved positions: 8; minimum length of a block: 10). The resulting codon alignments from all single-copy orthologs were then concatenated to one supergene for species phylogenetic analysis. RAxML (v8.2.12) was used to build the species phylogenetic tree with parameters: -f a -N 1000 -m PROT-GAMMAILGX, and r8s (v1.70)[65] was used to compute the mean substitution rates along each branch and estimate the species divergent time.

**Synteny analyses.** All-vs-all BLASTP searches (with an E value cutoff of $10^{-5}$) was performed to identify paralogous or orthologous gene pairs. Collinear blocks containing at least five genes were identified using MCScanX[66] with parameters: -s 5 -m 5. The Circos software (v0.69)[67] was used to illustrate the positional relationships among syntenic blocks and genomic features in the broomcorn millet genome.

**Calculation of 4DTv distance.** The transversion rates of fourfold generation sites (4DTv) between gene pairs located in syneny blocks were calculated using an in-house perl script. Tandemly duplicated genes that matched the same homeolog were only counted once.

**Gene copy number and phylogenetic analyses of genes.** In general, genes with defined functions in reference organisms were used as baits to search against the orthologous classification results from OrthoFinder (v1.1.4). For $C_4$ genes, foxtail genes involved in $C_4$ photosynthesis were used as baits. The copy numbers of genes were summarized manually based on their functions. To construct phylogenetic tree of NAD-ME and NADP-ME genes, the sequence from genomic regions that cover 5 kb upstream and downstream of the related genes were extracted and submitted to Fgenesh (http://www.softberry.com) for gene model correction. Based on the corrected gene models, codon alignment and phylogenetic tree construction was performed in MEGA7. The gene tree topology was then modified using treefix[68] and the branch length was calculated using RAxML (v8.2.12) with the following parameters: -f e –t gene.tree -m GTRGAMMA.

For BTB proteins, rice and *Arabidopsis* BTB proteins were used as baits to search against the proteins sequences of foxtail millet, broomcorn millet, rice and *Arabidopsis* with BLAST (E value cutoff 10). The resulting candidates were then submitted to the SMART (http://smart.embl-heidelberg.de) and pfam (https://pfam.xfam.org) database for domain architecture analyses. The conserved domains were identified with an E value cutoff of $10^{-5}$. The BTB proteins were divided into subgroups depending on the identity and position of other associated domains. To construct phylogenetic tree of BTB proteins, amino acid sequences of BTB domains were extracted and multiple sequence alignment was performed in MEGA7 and the tree was constructed using the Neighbor-Joining method with 1000 bootstraps.

**RNA extraction and transcriptome analyses.** Total RNAs from broomcorn millet tissues were isolated using the Plant RNeasy Mini Kit (Qiagen) following the manufacturer's instructions. RNA was eluted in 50 μL RNase-free water per reaction. Strand-specific mRNA libraries were prepared at Core Facility for Genomics at Shanghai Center for Plant Stress Biology (PSC) using NEBNext Ultra Directional RNA Library Prep Kit for Illumina (New England BioLabs, Cat No. E7420). The libraries were then sequenced on an HiSeq2500 (Illumina) using the paired-end 125-bp sequencing mode.

The adapter sequences and bases with a quality score lower than 30 were trimmed from raw sequencing reads. The clean reads were then mapped to Pm_0390_v1 assembly using subread-align (v1.5.1)[69]. Only uniquely mapped paired-end reads were retained for read counting for the annotated gene models. The count table and RPKM (reads per kb per million reads) were calculated using Gfold (v1.1.2)[70].

**Reporting summary.** Further information on experimental design is available in the Nature Research Reporting Summary linked to this article.

## Data availability

The genome assembly and sequence data for *P. miliaceum* was deposited at NCBI under BioProject number PRJNA431363. The genome assembly is also available through CoGe (Genome ID: 52484). The developmental transcriptome data were deposited at NCBI under BioProject number PRJNA431485. The source data underlying Figs. 1a–f, 2a–c, 3a, 4c and Supplementary Figs. 4, 9a, 9b, 10, 12 are

provided as a Source Data file. A reporting summary for this article is available as a Supplementary Information file.

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

## Acknowledgements
We thank Xingtan Zhang, Dipak Santra, Mingju Li, Shigang Wu, Jue Ruan for technical assistance; we thank Yuanyuan Li and Ray Ming for critical reading of the manuscript. Funding for this study was provided by the Chinese Academy of Sciences (CAS) to J.-K.Z. and Shanghai Science and Technology Committee (17391900200), Strategic Priority Research Program of CAS (XDB27040108), Youth Innovation Promotion Association CAS (2014242), National Key R&D Program of China (2016YFA0503200) and CAS to H.Z.

## Author contributions
Heng Z., J.-K.Z., P.S.S., J.C.S. designed the experiments, C.Z., D.M., D.L., L.X., S.R., P.D., W.J., R.H., M.Z., Y.C. performed the experiments, C.Z., L.L., D.L., Q.T., S.R., L.P., Y.S., J.H., X.F., F.L., Hui Z., B.F., X.Z., R.L., J.C.S., HengZ. analyzed data, Heng Z., J.-K.Z., C.Z., L.L., X.Z., J.C.S. wrote the paper.

## Additional information

**Competing interests:** The authors declare no competing interests.

