## [Peer Review File · Nature Communications]

Reviewers' comments:

Reviewer #1 (Remarks to the Author):

Zou et al have presented the genome of broomcorn millet (*Panicum miliaceum* L.). The study reports the assembly, annotation and comparative analysis of broomcorn millet with other sequenced grasses. The genome sequence might be a useful resource for breeding. However, the authors have presented a very routine analysis which significantly lacks validations. The C4 photosynthesis model shown in the MS is quite informative but there are no functional validations to support the model. The authors have used advanced technologies like Pacbio and Hi-C but still, the N50 is quite low which raises concerns about assembly. Either the authors should look into the assembly or give proper justification for this low N50.

Based on above, I am not very confident to recommend this MS for publication in a journal like Nature Communications, especially when the genomes of many members of Poaceae family have been decoded and their importance in agriculture has already been deciphered .

Also, I have the following suggestions:

1. L157 "We annotated repetitive sequences of the genome using both in silico prediction and homology-based approaches". Maybe the authors want to say de novo instead of in-silico
2. L131 Please check the number of contigs. If total contigs were 5,541 and the contigs assembled into pseudomolecules were 4,249. Then the unassigned contigs should be 1,292 instead of 1,291/
3. The consistency in MS needs to be improved to avoid confusion. For instance, in MS somewhere it is mentioned (E value cut off of 10) and at another place (E value cut off of 1e-5).
4. L690-691: "Plant DNeasy Mini Kit" should be "DNeasy Plant Mini Kit".
5. The authors can mention the pacbio sequencing chemistry, gene expansion analysis methodology, versions of different tools used in the methodology section.
6. The language of the MS needs to be considerably improved. There are many confusing sentences and grammatical mistakes in the MS.
6. The authors can also include %age of gaps in Table 1.
7. In Supplementary Table 12, the percentage of total annotated + total unannotated is > 100%
8. Figure 1: Figure and legend text do not match, d) is written as SNP density in the figure and in legend it is mentioned as SSR density.
9. L680-681: CGRIS's expansion seems incorrect.

Reviewer #2 (Remarks to the Author):

Revisions referring to my and other two reviewers' comments were well done. Especially, the most crucial point of this article, the method and resultant genome sequence quality were better delineated than that of the previous manuscript. The improvement of linkage map was excellent.

Reviewer #3 (Remarks to the Author):

In the revision by Zou et al., most of the issues raised by reviewers were addressed sufficiently. However, major assumptions in the C4 pathway analysis still remain but this is an interesting analysis that generates new questions.

As we consider this study primarily for resource value, we think it is critical to ensure the accuracy of the genome assembly and annotation. Specifically, we ask you to provide the access to the gene model data, which will be further reviewed by our previous reviewer(s). In addition, we note that the genomic DNA sequence generated in this study is missing in the current version. Reviewer #3 used the GenBank accession code provided in the previous version for dotplot analysis with foxtail millet genome. However, the result is significantly different from what you presented in Supplementary Fig 7b. Thus, we ask you to confirm the correct accession code in the revision and provide a reanalysis or an explanation of the differing dotplot analysis results. Further, we expect you to properly address the low N50 concern raised by Reviewer #1.

Response: We apologize for the confusion that Reviewer #3 has encountered. We have now updated our genome submission at NCBI database and the latest version of the genome is under the same BioProject number (PRJNA431363) and an updated genome version number (PQIB02000000). The data was deposited on Sep 20th and set to be released immediately, although it may take a little time for NCBI to process it. We also deposited the latest version of the genome to CoGe (<https://genomeevolution.org/coge/GenomeInfo.pl?gid=52484>).

The low N50 concern of Reviewer #1 is addressed below.

Reviewers' comments:

Reviewer #1 (Remarks to the Author):

Zou et al have presented the genome of broomcorn millet (*Panicum miliaceum* L.). The study reports the assembly, annotation and comparative analysis of broomcorn millet with other sequenced grasses. The genome sequence might be a useful resource for breeding. However, the authors have presented a very routine analysis which significantly lacks validations. The C4 photosynthesis model shown in the MS is quite informative but there are no functional validations to support the model. The authors have used advanced technologies like Pacbio and Hi-C but still, the N50 is quite low which raises concerns about assembly. Either the authors should look into the assembly or give proper justification for this low N50.

Response: Our data was generated 3 years ago using an earlier version of PacBio sequencer

(RS II). The read lengths were shorter than data generated using the latest PacBio technology. We believe that the relatively short subread N50 is the main reason for the relatively low N50 of our assembly. The Canu assembler used in our study is highly rated by the community; the original canu paper by Koren et al. has been cited 135 times since its publication in May 2017. In order to test whether increased subread N50 of the data can help with the assembly, we have tested our assembly algorithm using the PacBio data generated by Dr. Lai's group (produced with the latest PacBio model, Sequel). Our analyses indicate that while similar coverage of corrected reads were used for genome assembly (36.36x vs 37.34x), the read length from Dr. Lai's newer dataset is ~2-fold longer compared to the read length of our older data.

Next, we performed genome assembly on Dr. Lai's dataset using the same default parameters and generated a new assembly with contig N50 of 5.4 Mbp, which is equivalent to the contig N50 from their own analysis.

	Longmi4 statistics
# contigs (>= 1000 bp)	1187
Total length (>= 1000 bp) (bp)	853,211,571
Longest contig (bp)	22,168,111
Contig N50 (bp)	5,406,544

These analyses, together with other quality control metrics (Line 143-157 and Line 186-189), suggest that the relatively low contig N50 of our assembly is not an indicator of errors in the assembly.

Based on above, I am not very confident to recommend this MS for publication in a journal like Nature Communications, especially when the genomes of many members of Poaceae family have been decoded and their importance in agriculture has already been deciphered .

Also, I have the following suggestions:

1. L157 "We annotated repetitive sequences of the genome using both in silico prediction and homology-based approaches". Maybe the authors want to say de novo instead of in-silico

Response: In many cases “*in silico*” and “*de novo*” can be used interchangeably for gene model prediction, but “*in silico*” is more commonly used in the literature. Searching for the phrases “*in silico* prediction” and “*de novo* predication” at Pubmed generated 1374 and 123 papers respectively (Sep 12th, 2018).

2. L131 Please check the number of contigs. If total contigs were 5,541 and the contigs assembled into pseudomolecules were 4,249. Then the unassigned contigs should be 1,292 instead of 1,291/

Response: The number of contigs assembled into pseudomolecules should be 4,250. This has been corrected.

3. The consistency in MS needs to be improved to avoid confusion. For instance, in MS somewhere it is mentioned (E value cut off of 10) and at another place (E value cut off of 1e-5).

Response: Different E value cut offs were used in two different analyses. In the case where a looser cut off was used, domain prediction was later performed to make sure only the proteins of interest were retained for further analyses.

4. L690-691: "Plant DNeasy Mini Kit" should be "DNeasy Plant Mini Kit".

Response: Corrected.

5. The authors can mention the pacbio sequencing chemistry, gene expansion analysis methodology, versions of different tools used in the methodology section.

Response: The information was added to the “Materials and Methods” section.

6. The language of the MS needs to be considerably improved. There are many confusing sentences and grammatical mistakes in the MS.

Response: The manuscript has been edited by a native English editor.

6. The authors can also include %age of gaps in Table 1.

Response: Corrected. Stats on “Undetermined bases” were added to Table 1.

7. In Supplementary Table 12, the percentage of total annotated + total unannotated is > 100%

Response: Corrected.

8. Figure 1: Figure and legend text do not match, d) is written as SNP density in the figure and in legend it is mentioned as SSR density.

Response: The figure legend was corrected to “SNP density”.

9. L680-681: CGRIS's expansion seems incorrect.

Response: The CGRIS short name is removed.

Reviewer #2 (Remarks to the Author):

Revisions referring to my and other two reviewers' comments were well done. Especially, the most crucial point of this article, the method and resultant genome sequence quality were better delineated than that of the previous manuscript. The improvement of linkage map was excellent.

Response: We would like to thank the reviewer for the positive comments.

Reviewer #3 (Remarks to the Author):

In the revision by Zou et al., most of the issues raised by reviewers were addressed sufficiently. However, major assumptions in the C4 pathway analysis still remain but

this is an interesting analysis that generates new questions.

Response: We are glad to know that the reviewer agreed that most of the issues have been sufficiently addressed. We agree that further validation of the C₄ photosynthesis model based on transcriptional and phylogenetic analyses requires more experiments, which we are carrying out right now.

REVIEWERS' COMMENTS:

Reviewer #3 (Remarks to the Author):

The authors have addressed my concerns.

REVIEWERS' COMMENTS:

Reviewer #3 (Remarks to the Author):

The authors have addressed my concerns.

Response: We thank the reviewer for the positive comment.